# Risk of Acute Myocardial Infarction Among New Users of Allopurinol According to Serum Urate Level: A Nested Case-Control Study

**DOI:** 10.3390/jcm8122150

**Published:** 2019-12-05

**Authors:** Sara Rodríguez-Martín, Francisco J. de Abajo, Miguel Gil, Diana González-Bermejo, Antonio Rodríguez-Miguel, Diana Barreira-Hernández, Ramón Mazzucchelli, Alberto García-Lledó, Luis A. García-Rodríguez

**Affiliations:** 1Clinical Pharmacology Unit, University Hospital Príncipe de Asturias, Alcalá de Henares, 28805 Madrid, Spain; rodriguezmartin.sara@gmail.com (S.R.-M.); antonio.hupa@gmail.com (A.R.-M.); dbarreirahdez@gmail.com (D.B.-H.); 2Department of Biomedical Sciences (Pharmacology Sector), University of Alcalá (IRYCIS), Alcalá de Henares, 28805 Madrid, Spain; 3Division of Pharmacoepidemiology and Pharmacovigilance, Spanish Agency for Medicines and Medical Devices (AEMPS), 28022 Madrid, Spain; mgilg@aemps.es (M.G.); dgonzalezb@aemps.es (D.G.-B.); 4Department of Rheumatology, University Hospital Fundación Alcorcón, 28922 Alcorcón, Madrid, Spain; RMazzucchelli@fhalcorcon.es; 5Department of Cardiology, University Hospital Príncipe de Asturias, Alcalá de Henares, 28805 Madrid, Spain; josealberto.garcia@salud.madrid.org; 6Department of Medicine, University of Alcalá, Alcalá de Henares, 28805 Madrid, Spain; 7Spanish Center for Pharmacoepidemiologic Research (CEIFE), 28004 Madrid, Spain; lagarcia@ceife.es

**Keywords:** allopurinol, serum uric acid levels, acute myocardial infarction, hyperuricemia, gout

## Abstract

Objectives: To test the hypothesis that allopurinol reduces the risk of acute myocardial infarction (AMI) in hyperuricemic patients and to assess whether the effect is dependent on dose, duration and serum uric acid (SUA) level attained after treatment. Methods: Nested case-control study over the period 2002–2015. From a cohort of patients aged 40–99 years old, we identified incident AMI cases and randomly selected five controls per case, matched for exact age, sex and index date. Adjusted odds ratios (AOR) and 95% CI were computed through unconditional logistic regression. Only new users of allopurinol were considered. Results: A total of 4697 AMI cases and 18,919 controls were included. Allopurinol use was associated with a reduced risk of AMI mainly driven by duration of treatment (AOR ≥180 days = 0.71; 95% CI: 0.60–0.84). Among long-term users (≥180 days), the reduced risk was only observed when the SUA level attained was below 7 mg/dL (AOR_<6 mg/dL_ = 0.64; 95% CI: 0.49–0.82; AOR_6–7mg/dL_ = 0.64; 95%CI:0.48-0.84); AOR_>7mg/dL_ = 1.04; 95% CI: 0.75–1.46; *p* for trend = 0.001). A dose-effect was observed but faded out once adjusted for the SUA level attained. The reduced risk of AMI occurred in both patients with gout and patients with asymptomatic hyperuricemia. Conclusions: The results confirm a cardioprotective effect of allopurinol which is strongly dependent on duration and SUA level attained after treatment.

## 1. Introduction

Mounting evidence shows that hyperuricemia and gout are associated with an increased risk of cardiovascular disease [1,2,3,4], suggesting that they are important risk factors, though a reverse causality cannot be ruled out [5]. In this context, it is of interest to know whether urate-lowering therapy (ULT) reduces the risk of cardiovascular events in hyperuricemic patients. Allopurinol, a xanthine oxidase inhibitor (XOI), is the first-line ULT for patients with gout to prevent acute flares [6,7]. Often, it is also used to treat asymptomatic hyperuricemia, though this practice is generally not supported [8,9]. In the last few years, many studies have provided evidence on the cardiovascular benefits of allopurinol [5,10]. In 2015, our group reported a risk reduction of acute myocardial infarction (AMI), which was dose- and duration-dependent, among allopurinol initiators [11], but recent studies yielded conflicting results, with some reporting a protective effect [12,13,14,15] and others no effect [16,17,18] or an increased risk [19]. Thus, the subject is still a matter of controversy and further studies are needed [1,2].

Another important issue still unsolved is whether the potential protective effect of allopurinol on AMI is related or not with the lowering of serum uric acid (SUA) levels [1,2,20]. Previous studies either did not address the issue [12,14] or did not find a relation [11,13,15,16], although the sample size could have been a limitation to draw a firm conclusion. 

In the present study, performed in hyperuricemic patients with or without gout, we aimed to confirm the protective effect of allopurinol on AMI and to assess the role of daily dose, duration of treatment and SUA level attained after treatment.

## 2. Patients and Methods

### 2.1. Data source and Study Design

We conducted a case-control study nested in a primary cohort selected from BIFAP (a Spanish primary healthcare database; see Appendix A for details) [21] over the study period 1 January 2002 to 31 December 2015. The primary cohort was composed of patients aged 40 to 99 years old, registered with their primary care physician (PCP) for at least 1 year and who did not have a previous record of cancer or AMI. The first day the patients met all the criteria mentioned above was the “start date” of the follow-up. The primary cohort was composed of 3,764,470 subjects. They were then followed until the first of the following events: an incident AMI, 100 years old, a record of cancer, death, or the end of the study period. 

### 2.2. Selection of Cases and Controls

Incident AMI cases were initially searched through both codes and text on diagnosis fields and validated through manual review of clinical records (see Appendix B for details). The “index date” was considered the date of the first record of AMI. Five controls per case were randomly selected from the underlying cohort following a risk set sampling in which controls were individually matched to cases by exact age, sex, and index date.

### 2.3. Definitions of Gout and Asymptomatic Hyperuricemia

Once cases and controls were selected, we identified those having gout or asymptomatic hyperuricemia. A patient was considered to have “gout” when his/her automated clinical record had a specific code of gout or related terms within the diagnosis field (“gout”, “tophus”, “gouty arthritis” or “podagra”). A patient was considered to have “asymptomatic hyperuricemia” when, having no record of gout, he/she had one of the following criteria (in descending order): (1) a text of “hyperuricemia” within the diagnosis field; (2) at least one SUA level prior to the index date above 8 mg/dL (480 μm/L) in men or 7 mg/dL (420 μm/L) in women; (3) at least two SUA levels prior to the index date between 7 and 8 mg/dL in men or between 6 and 7 mg/dL in women. Patients with an isolated diagnosis of uric renal lithiasis were not considered.

### 2.4. New Users Design

The analysis was performed among new-users of allopurinol. For that purpose, we excluded all cases and controls with a recorded prescription of allopurinol before the start date [22] (Figure 1).

### 2.5. Exposure Definition

Patients were classified as “current users” of allopurinol when the last prescription ended within 30 days prior to the index date; “recent users” when ended between 31 and 365 days prior to the index date; “past users” when ended more than 365 days prior to the index date; and “non-users” when there was no recorded prescription of allopurinol ever before index date.

The effect of daily dose was studied among current users of allopurinol. Two categories were considered: “low-dose” when it was less than 300 mg and “high dose” when it was equal or higher than 300 mg (most patients of this category were prescribed a dose of 300 mg/day and only three patients were on daily doses over 300 mg). Duration of treatment was computed summing up consecutive prescriptions (with a maximum gap of 90 days between the end of one and the starting of the next), and then grouped in two categories: less than 180 days and 180 days or longer.

### 2.6. Serum Uric Acid (SUA) Levels Attained

Among allopurinol current users, we identified the latest SUA level recorded beyond 28 days after treatment initiation and before the index date. We grouped patients in three categories (<6 mg/dL; 6 to 7 mg/dL and >7 mg/dL) (<360 μm/L, 360μm/L to 420μm/L and >420 μm/L) and examined the existence of a trend. We used two criteria to set the categories of SUA levels: (1) the target recommended in American and European guidelines for the ULT treatment in patients with gout (<6 mg(dl) [6,7]; and (2) the limit of solubility for uric acid in plasma (7 mg/dL) [20].

### 2.7. Potential Confounding Factors

The following comorbidities (recorded before the index date) were assessed as potential confounding factors: cerebrovascular disease (ischemic, hemorrhagic or non-specified stroke and transient ischemic attack), heart failure, angina pectoris (recorded as such and/or use of nitrates), peripheral artery disease (PAD), hypertension, diabetes (recorded as such and/or use of glucose-lowering drugs), dyslipidemia (recorded as such and/or use of lipid-lowering drugs), rheumatoid arthritis, osteoarthritis, and chronic kidney disease. In addition, we considered the following factors: number of visits to the PCPs in the year prior to the index date, body mass index (BMI), smoking, and current use of the following drugs: low-dose aspirin, non-aspirin antiplatelet drugs, oral anticoagulants, non-steroidal anti-inflammatory drugs (NSAIDs), colchicine, corticosteroids, angiotensin-converting enzyme inhibitors (ACEI), angiotensin II receptor blockers (ARB), calcium-channel blockers, beta-blockers, alfa-blockers, and diuretics. 

### 2.8. Statistical Analysis

The association between incident AMI and the exposure to the drugs of interest was evaluated by computing the Odds Ratio (OR) and the corresponding 95% confidence intervals (CI) through unconditional logistic regression models. First, we estimated the crude ORs including only the exposure and the matching variables (age, sex and calendar year); then, we computed the adjusted OR (AOR) adding all the potential confounding factors mentioned in the previous section. Furthermore, we studied the interaction with age (stratified as less than 65 and equal or greater than 65 years old), sex, obesity (defined as a BMI over 30 kg/m^2^), history of diabetes, history of atherothrombotic disease (includes angina pectoris, PAD or cerebrovascular accident), and concomitant use of statins (all of them patients with dyslipidemia) or drugs inhibiting the renin-angiotensin system (either ACEI or ARB). For the statistical evaluation of the interaction, we ran adjusted models across different categories of the interacting variables and computed the AORs associated with current use of the drugs of interest as compared to non-use in each stratum. The AORs across strata were compared using the test of interaction described by Altman and Bland [23]. Results were considered statistically significant when the p-value was lower than 0.05.

Missing values for smoking (34.7%), BMI (20.1%) and SUA level (12.1%) were addressed performing multiple imputations by chained equation (MICE) models [24] (see Appendix C for details).

We conducted all analyses using STATA version 15/SE (StataCorp. College Station, TX, USA).

### 2.9. Sensitivity Analyses

Two sensitivity analyses were performed: (1) using only patients with complete data of SUA levels; and (2) excluding hypertension from the model, as it may be acting as an intermediate variable in the causal pathway between gout/hyperuricemia and AMI [25].

### 2.10. Ethics Review

Access to anonymized data from BIFAP was granted by the BIFAP Scientific Committee (project #04/2016; approval date: 26 May 2016). According to the Spanish law, no specific ethical review was required for studies using fully anonymized data.

## 3. Results

We included a total of 4697 incident AMI cases and 18,919 controls, all of them with a record of either gout or asymptomatic hyperuricemia (Figure 1). Their characteristics are outlined in Table 1. As expected, the prevalence of cardiovascular risk factors and use of cardiovascular drugs was higher in cases as compared to controls.

### 3.1. Allopurinol Use and Risk of AMI and Effect of Dose and Duration

Current use of allopurinol was slightly lower among cases (321; 6.8%) than in controls (1417; 7.5%), leading to a crude OR of 0.93 (0.82–1.05). After full adjustment, the AOR went down to 0.84 (95% CI: 0.73–0.96) and decreased even further at a dose of 300 mg or higher (AOR = 0.75; 95% CI: 0.60-0.93), after durations of 180 days or longer (AOR = 0.71; 95% CI: 0.60–0.84) (test for trend, *p* = 0.0001) and, particularly, when these two conditions were met (AOR = 0.61; 95% CI: 0.46–0.81). The strongest reduction of AMI risk was observed when patients used high doses for 2 years or longer (AOR = 0.48; 95% CI: 0.31–0.75). A reduction of risk was still observed with low daily doses of allopurinol when they used it long-term (AOR = 0.77; 95% CI: 0.63–0.94) (Table 2).

### 3.2. Allopurinol Use and Risk of AMI According to SUA Levels Attained

We found a statistically significant reduced risk of AMI when the SUA level was below 6 mg/dL (AOR = 0.77; 95% CI: 0.63–0.96) (Figure 2). At durations longer than 180 days, a reduced risk was observed when the SUA level was either less than 6 mg/dL (AOR = 0.64; 95% CI: 0.49–0.82) or between 6 and 7 mg/dL (AOR = 0.64; 95% CI: 0.48–0.84), but not when the SUA level was higher than 7 mg/dL (AOR = 1.04; 95%: 0.75–1.46) (test for trend, *p* = 0.0001) (Figure 2). Daily dose barely modulated the AORs associated with allopurinol within the different SUA level categories (Figure 2).

### 3.3. Allopurinol Use and AMI Stratified by Gout and Asymptomatic Hyperuricemia

In gout patients, the current use of allopurinol was associated with a decreased risk of AMI when used at high doses (AOR = 0.66; 95% CI: 0.48–0.90) or for long-term periods (AOR = 0.68; 95% CI: 0.53–0.89) (test for trend, *p* = 0.004), and, particularly, when patients met both features (AOR = 0.49; 95% CI: 0.33–0.73). In long-term users, a trend with SUA level was observed (*p* = 0.001) (Figure 3). Among asymptomatic hyperuricemic patients, the effect of allopurinol appeared to be weaker, but reached statistical significance when used for periods of 180 days or longer (AOR = 0.75; 0.59–0.94) (test for trend, *p* = 0.010), and particularly when long durations were associated with SUA levels below 7 mg/dL (AOR_<6 mg/dL_ = 0.66; 0.46–0.95; AOR_6-7mg/dL_ = 0.63; 0.43–0.94; AOR_>7mg/dL_ = 1.31; 0.81–2.13; *p* for trend = 0.001) (Figure 4).

### 3.4. Allopurinol Use and Risk of AMI in Different Subgroups

The results of the potential interaction of allopurinol use with sex, age, obesity, antecedents of atherothrombotic disease, diabetes, and concurrent use of statins or drugs blocking the renin-angiotensin system are shown in Figure 5. No evidence of a statistical interaction was found with any of them.

### 3.5. Sensitivity Analyses

(1) In patients with complete SUA data, we also found a reduced risk of AMI associated with lower SUA levels among allopurinol current users (AOR_<6 mg/dL_ = 0.70; 95% CI: 0.53–0.91; AOR_6–7 mg/dL_ = 0.88; 95%CI:0.63–1.24; AOR_>7 mg/dL_ = 1.03; 95%CI:0.76–1.40) showing a significant trend (*p* = 0.011) (Table 3); (2) the exclusion of hypertension from the adjusted model had no impact in the AOR estimates.

## 4. Discussion

In the present study, we found that the use of allopurinol was associated with a reduction of AMI risk, which was mainly observed when the duration of treatment was 180 days or longer and when the SUA level attained after treatment was below 7 mg/dL. A significant dose effect was also observed but proved less relevant once results were adjusted for the SUA level. Such cardioprotective effect of allopurinol was found in both gout and asymptomatic hyperuricemic patients. 

The present study confirms the results we previously reported on allopurinol and AMI risk [11], but with more robust data. So far, other three observational studies performed in different countries (France [12], Denmark [13] and the US [14]),with different populations (either general population [12], hyperuricemic subjects [13], and patients with gout and diabetes [14]) and using different designs (case-control [12] and cohort [13,14] studies) have consistently reported a protective effect of allopurinol on atherothrombotic events (AMI alone [12], AMI or stroke [14], AMI or stroke or cardiovascular death [13]) ranging from 11% [13] to 33% [14]. Also, Wei et al. [26], comparing high doses with low doses of allopurinol, found a hazard ratio of 0.69 (95% CI: 0.50–0.94) for cardiovascular events in the UK. Finally, Bredemeier et al. [27], in a meta-analysis of randomized clinical trials, estimated a relative risk of AMI of 0.38 (95% CI: 0.17–0.83) among allopurinol users as compared to placebo or no treatment. Other authors [16,17,18], however, did not find a reduced risk, although the very low doses used by patients [18] and the low adherence to treatment observed [17] could partly explain some of the discrepant results.

The main novelty of the present study is the close relation found between AMI risk and SUA level reached after allopurinol treatment. Among long-term users, the reduced risk was around 40% when SUA level was below 7 mg/dL, while no benefit was observed when SUA level was over 7 mg/dL. These results are apparently in contrast with those by Desai et al. [28], who did not find a decreased risk of cardiovascular events with a reduction of >3 mg/dL in SUA levels. Nevertheless, they did not evaluate specifically the effect of allopurinol (actually, around 40% patients were not on ULT) and, hence, results are not comparable. Furthermore, the intensity of reduction may be less important than reaching a SUA level below a certain threshold. It is interesting to note that 7 mg/dL is close to the accepted solubility threshold of uric acid (6.8 mg/dL or 404.5 μmol/L) [20] and our data suggest that this level may be critical to obtain cardiovascular benefits. Recently, Pérez-Ruiz et al. [29] reported an increased risk of all-cause death and cardiovascular death in gout patients who failed to reach a target SUA level <360 μmol/L (6 mg/dL).

The underlying biological mechanism for allopurinol cardioprotection is unknown, although most hypotheses link the benefits of allopurinol with the inhibition of xanthine oxidase and the resulting reduction of reactive oxygen species (ROS) [30]. In a series of elegant clinical studies, Struthers et al. [31,32] provided compelling evidence of a remarkable effect of allopurinol on oxidative stress when used at very high doses (600 mg), which resulted in an improvement of the endothelial dysfunction. Our data show that the cardioprotective effect of allopurinol is stronger at higher doses which is compatible with this hypothesis, although such dose-relation tends to fade out when results are adjusted for the SUA level attained. However, it is important to note that in our study, only three patients used allopurinol at doses higher than 300 mg which might have prevented us from observing a relevant direct effect independent from the SUA level attained. The data show that the cardioprotective effect of allopurinol improves as duration of treatment increases, which is consistent with results from other studies [11,12]. Both the inflammatory state induced by urate deposits [33] (and maybe soluble uric acid itself [34]), and the atherosclerotic process caused by endothelial dysfunction and other cardiovascular risk factors associated with gout/hyperuricemia [20,33], require time to be reversed and this could be the biological explanation for the time-dependent effect of allopurinol. Additionally, the start of treatment with ULT mobilizes the urate deposits which could increase the pro-inflammatory state and, in turn, the cardiovascular risk, a factor which may counteract any potential short-term protective effect of allopurinol.

Taking all these data together, we postulate that the protective effect of allopurinol on AMI may have a double component: a direct effect on ROS only observable with high daily doses (300 mg or higher); and a SUA-dependent component, clearly evident when serum concentrations are maintained below 7 mg/dL. Nevertheless, both mechanisms are closely linked as SUA level reduction is also a marker of the inhibition of XO activity. The present study only focused on AMI, but it is expected that a similar effect could be observed on other atherothrombotic events like ischemic stroke. 

The reduced risk of AMI with allopurinol use was particularly observed in patients with gout, but it is noteworthy that we also observed a protective effect in patients with asymptomatic hyperuricemia provided that treatment was prolonged and SUA levels maintained below 7 mg/dL. Whether the protective effect observed in these patients should change the general recommendation of not using allopurinol in them [6,7,8,9], is a question beyond the scope of the present research and should be addressed in specific studies considering the risks of allopurinol, in particular severe cutaneous reactions [35,36,37,38].

We did not find a statistically significant interaction between allopurinol use and any of the variables examined, and we must conclude that allopurinol elicits similar benefits in all subgroups studied. In the case of use of statins or ACEI/ARB, it means that the benefits of allopurinol are added to the cardiovascular benefits of these drugs. A lower relative effect was found in the subgroups with the highest cardiovascular risk, but this is an expected result as the same absolute risk reduction translates into lower and lower relative risks when the background risk increases. 

In our study population of gout patients, the risk of AMI was roughly estimated in two per 1000 person-years. Assuming that the use of allopurinol was associated with a long-term risk reduction of around 40% (when SUA level is below 7 mg/dL), the risk of AMI among exposed would be 1.2 per 1000 person-years, leading to an absolute risk reduction (ARR) of 0.8 per 1000 person-years and a Number Needed to Treat (NNT = 1/ARR) of 1250 persons per year. In other words, for every set of 1250 patients treated with allopurinol per year, one patient would be saved from having an AMI attributable to this drug.

Our study has a number of strengths: (1) researchers who performed the case validation were blinded to drug exposure which prevents a differential misclassification of the disease conditioned by the exposure; (2) controls were randomly extracted from the underlying cohort, assuring that they represented the exposure of the population that gave rise to cases and thereby avoiding a selection bias [39]; (3) all patients were hyperuricemic which makes the groups more comparable in terms of background cardiovascular risk than those studies performed in the general population; (4) only new users of allopurinol were considered, which prevents from bias linked to early events [22]; and (5) the sensitivity analyses did not materially change the results, showing that they are robust. 

The main limitations of the study are as follows: (1) It is an observational study, so residual confounding due to unknown or unmeasured factors is still possible; (2) SUA levels were not available in 12.1% of patients; however, we applied a multiple imputation method which is recognized as the best method to address missing data [40]; on the other hand, the sensitivity analysis including only patients with complete data showed consistent results, though less precise; (3) physicians fill prescriptions through the computer, so a misclassification of the exposure for not recording prescription-only drugs is unlikely, but the adherence to treatment by patients cannot be assured; nevertheless, the impact of this potential error would have operated against the working hypothesis; (4) the primary care physicians do not record reliable information on diet and other lifestyle factors and the effect of a modification of these factors on SUA levels could not be assessed; however, guidelines agree on stating that these interventions have little effect on urate concentrations [6] and thus the impact of this missing information is probably small; and (5) the exposure to other ULT drugs different from allopurinol was very small to perform a meaningful analysis with them (febuxostat and benzbromarone, the only ULT drugs available in Spain during the study period, had a very limited use: only 14 cases and 34 controls were current users of febuxostat; and no cases and only three controls were current users of benzbromarone).

## 5. Conclusions

The results of the present study show a protective effect of allopurinol on AMI risk, which is strongly dependent on the duration and SUA level attained after treatment. The greatest protective effect is observed when the duration of treatment is 6 months or longer and SUA level is below 7 mg/dL. Once these conditions are met, the daily dose (up to 300 mg) seems to play a minor role. The cardiovascular benefits of allopurinol should be an extra incentive for physicians and patients to be fully adherent to ULT in the long-lasting treatment of gout, in addition to the prevention of acute flares. Although our data suggests that asymptomatic hyperuricemic patients might also gain cardiovascular benefits with allopurinol, the benefit–risk ratio in this population should be further evaluated in specific studies.

## Reference

## Figures and Tables

**Figure 1 jcm-08-02150-f001:**
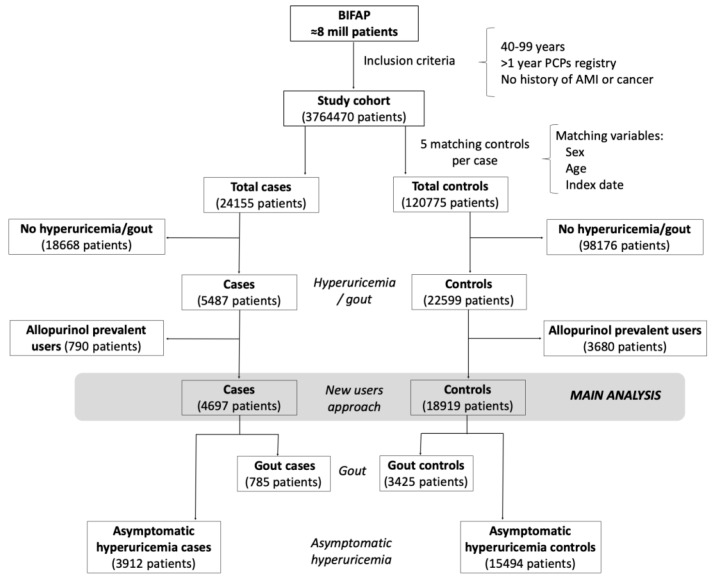
Flowchart of patient selection.

**Figure 2 jcm-08-02150-f002:**
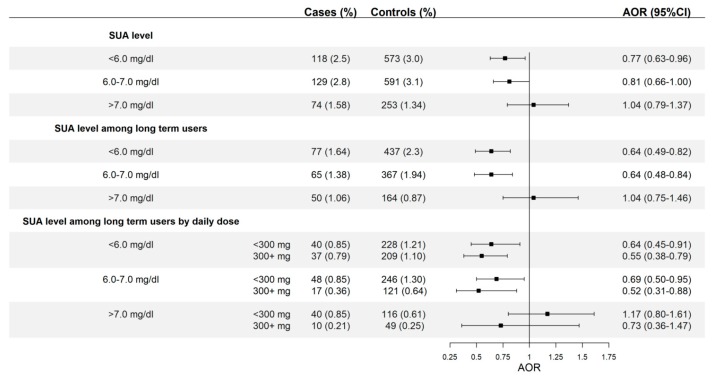
Risk of acute myocardial infarction among current users of allopurinol by serum uric acid (SUA) level in different scenarios. Long-term use is defined as a duration of 180 days or longer. AOR: Adjusted Odds Ratio, 300+ mg means 300 mg or higher.

**Figure 3 jcm-08-02150-f003:**
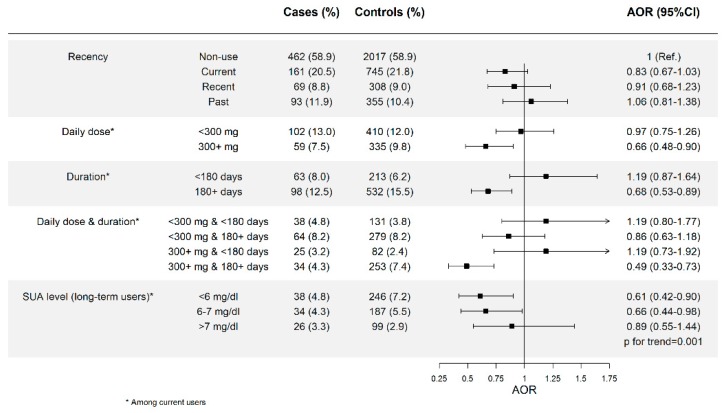
Patients with gout only. Risk of AMI among new users of allopurinol according to recency of use, daily dose, duration of treatment, and SUA level reached after treatment. AOR: Adjusted Odds Ratio, 300+ mg means 300 mg or higher, 180+ days means 180 days or longer.

**Figure 4 jcm-08-02150-f004:**
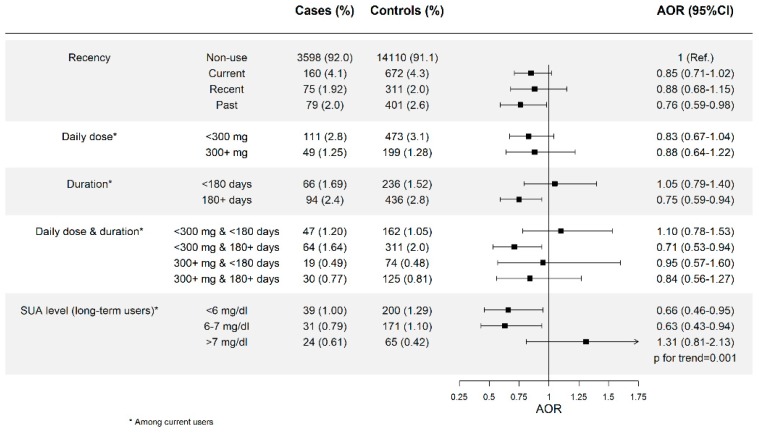
Patients with asymptomatic hyperuricemia. Risk of AMI among new users of allopurinol according to recency of use, daily dose, duration of treatment, and serum uric acid (SUA) level attained after treatment. AOR: Adjusted Odds Ratio, 300+ mg means 300 mg or higher, 180+ days means 180 days or longer.

**Figure 5 jcm-08-02150-f005:**
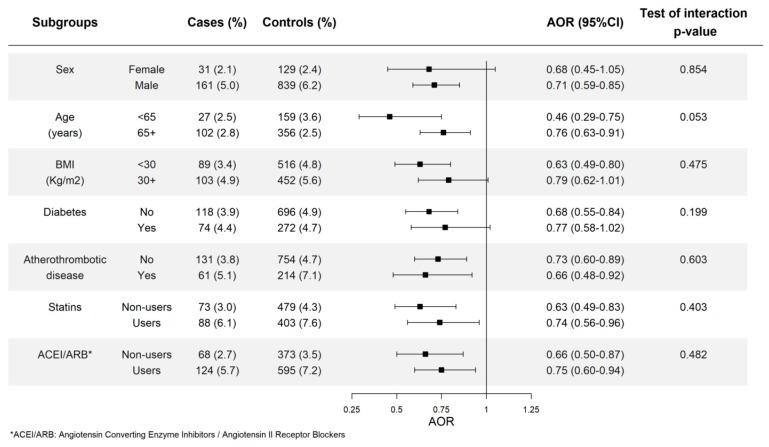
Assessment of interaction between long-term current use of allopurinol and different factors. Long-term use is defined as a duration of 180 days or longer. AOR: Adjusted Odds Ratio, BMI: Body mass index, Age 65+ means 65 years or older, BMI 30+ means 30 Kg/m^2^ or higher.

**Table 1 jcm-08-02150-t001:** Characteristics of cases and controls. All patients were hyperuricemic. Prevalent users of allopurinol were excluded.

	Cases *n* = 4697	Controls *n* = 18919	Crude OR * (95% CI)	Adjuster OR ^‡^ (95% CI)
Age, mean (SD)	70.0 (±12.9)	69.9 (±12.6)	-	-
Men	3199 (68.1)	13503 (71.4)	-	-
Visits (last 12 months)				
Up to 5	682 (14.5)	3819 (20.2)	1 (ref.)	1 (ref.)
6–15	1883 (40.1)	8127 (43.0)	1.35 (1.22–1.49)	1.21 (1.09–1.34)
16–24	1097 (23.4)	3892 (20.6)	1.69 (1.51–1.89)	1.41 (1.25–1.60)
≥25	1035 (22.0)	3081 (16.3)	2.06 (1.83–2.31)	1.58 (1.39–1.81)
BMI kg/m^2^				
Up to 24.9	434 (9.2)	1765 (9.3)	1 (ref.)	1 (ref.)
25–29 30–34	2158 (45.9) 1606 (34.2)	9097 (48.1) 6261 (33.1)	0.98 (0.87–1.10) 1.05 (0.93–1.18)	1.01 (0.90–1.14) 1.04 (0.92–1.18)
35–49	376 (8.0)	1413 (7.5)	1.06 (0.91–1.24)	0.96 (0.82–1.13)
≥40	123 (2.6)	383 (2.0)	1.26 (1.00–1.58)	1.10 (0.86–1.39)
Smoking				
Never smoking	1499 (31.9)	6792 (35.9)	1 (Ref.)	1 (Ref.)
Current smoker	1281 (27.3)	3637 (19.2)	1.45 (1.33–1.58)	1.43 (1.31–1.57)
Past smoker	394 (8.4)	1815 (9.6)	1.03 (0.91–1.17)	0.98 (0.86–1.11)
Unknown	1523 (32.4)	6675 (35.3)	Imputed	Imputed
CVA				
Ischemic	163 (3.5)	474 (2.5)	1.42 (1.19–1.71)	1.06 (0.87–1.29)
Hemorrhagic	17 (0.36)	68 (0.36)	1.03 (0.61–1.76)	0.89 (0.52–1.54)
Unspecified	112 (2.4)	404 (2.1)	1.16 (0.93–1.43)	0.89 (0.71–1.11)
TIA	150 (3.2)	495 (2.6)	1.25 (1.03–1.50)	1.03 (0.84–1.25)
Heart failure	338 (7.2)	957 (5.1)	1.45 (1.27–1.66)	1.22 (1.06–1.41)
Angina pectoris ^§^	660 (14.1)	1293 (6.8)	2.28 (2.06–2.52)	1.69 (1.51–1.90)
PAD	301 (6.4)	611 (3.2)	2.16 (1.87–2.50)	1.54 (1.32–1.79)
Hypertension	3444 (73.3)	13007 (68.8)	1.25 (1.16–1.35)	1.06 (0.97–1.17)
Diabetes ^||^	1669 (35.5)	4781 (25.3)	1.63 (1.52–1.75)	1.37 (1.27–1.47)
Dyslipidemia **	2847 (60.6)	9919 (52.4)	1.39 (1.30–1.49)	1.12 (1.04–1.20)
Rheumatoid arthritis	39 (0.83)	128 (0.68)	1.20 (0.84–1.72)	1.00 (1.69–1.46)
Osteoarthritis	601 (12.8)	2298 (12.2)	1.04 (0.94–1.14)	1.02 (0.92–1.13)
Chronic kidney failure	441 (9.4)	1389 (7.3)	1.32 (1.17–1.48)	1.08 (0.96–1.22)
Current use of Low-dose aspirin	969 (20.6)	2826 (14.9)	1.63 (1.49–1.77)	1.13 (1.03–1.25)
Other antiplatelet drugs	323 (6.9)	624 (3.3)	2.27 (1.97–2.61)	1.50 (1.28–1.76)
Oral anticoagulants	316 (6.7)	1405 (7.4)	0.90 (0.79–1.02)	0.70 (0.60–0.81)
NSAIDs	501 (10.7)	2005 (10.6)	1.04 (0.93–1.17)	0.94 (0.84–1.07)
Colchicine	67 (1.43)	220 (1.16)	1.28 (0.97–1.68)	1.22 (0.92–1.63)
Corticosteroids	124 (2.6)	333 (1.8)	1.55 (1.26–1.91)	1.38 (1.11–1.72)
ACE inhibitors	1187 (25.3)	4532 (24.0)	1.16 (1.07–1.25)	0.93 (0.85–1.02)
ARB	1089 (23.2)	3999 (21.1)	1.18 (1.09–1.28)	0.93 (0.85–1.03)
CCB	975 (20.8)	2963 (15.7)	1.51 (1.38–1.64)	1.23 (1.12–1.35)
Beta-Blockers	697 (14.8)	2136 (11.3)	1.41 (1.28–1.55)	1.09 (0.98–1.21)
Alfa-Blockers	183 (3.9)	706 (3.7)	1.08 (0.91–1.27)	0.85 (0.71–1.01)
Diuretics	1112 (23.7)	3937 (20.8)	1.23 (1.13–1.34)	1.00 (0.91–1.10))

Abbreviations: OR: Odds Ratio, SD: Standard Deviation, CI: Confident Interval. CVA: Cerebrovascular Accident, TIA: Transient Ischemic Accident, PAD: Peripheral Artery Disease, COPD: Chronic Obstructive Pulmonary Disease, BMI: Body Max Index, NSAIDs: Non-steroidal Anti-inflammatory Drugs, ACE: Angiotensin Converting Enzyme. ARB: Angiotensin II-Receptor Blockers; CCB: Calcium-channel blockers. * Adjusted only for the matching factors (age, sex and calendar year). ^‡^ Adjusted for the matching factors (age, sex and calendar year) and all covariables included in the table. ^§^ Recorded as such or when patients were using nitrates. ^||^ Recorded as such or when patients were using glucose-lowering drugs. ** Recorded as such or when patients were using lipid-lowering drugs.

**Table 2 jcm-08-02150-t002:** Risk of AMI among new users of allopurinol according to recency of use, daily dose and duration of treatment.

	Cases (%) *n* = 4697	Controls (%) *n* = 18919	Crude OR ^†^ (95% CI)	Adjusted OR ^‡^ (95% CI)
Recency				
Non-use	4060 (86.4)	16127 (85.2)	1 (Ref.)	1 (Ref.)
Current	321 (6.8)	1417 (7.5)	0.93 (0.82–1.05)	0.84 (0.73–0.96)
Recent	144 (3.1)	619 (3.3)	0.95 (0.79–1.14)	0.89 (0.73–1.08)
Past	172 (3.7)	756 (4.0)	0.93 (0.78–1.10)	0.89 (0.75–1.07)
Daily dose *				
<300 mg	213 (4.5)	883 (4.7)	0.98 (0.84–1.15)	0.90 (0.76–1.05)
≥300 mg	108 (2.3)	534 (2.8)	0.83 (0.68–1.03)	0.75 (0.60–0.93)
Duration *				
<180 days	129 (2.8)	449 (2.4)	1.17 (0.96–1.43)	1.13 (0.91–1.39)
≥180 days	192 (4.1)	968 (5.1)	0.81 (0.69–0.95)	0.71 (0.60–0.84)
180–729 days	116 (2.5)	537 (2.8)	0.88 (0.72–1.08)	0.76 (0.61–0.94)
>729 days	76 (1.62)	431 (2.3)	0.72 (0.57–0.93)	0.64 (0.50–0.83)
				*p* for trend = 0.0001
Daily dose & duration *				
<300 mg				
<180 days	85 (1.81)	293 (1.55)	1.18 (0.92–1.50)	1.15 (0.89–1.48)
≥180 days	128 (2.7)	590 (3.1)	0.88 (0.73–1.07)	0.77 (0.63–0.94)
180–729 days	76 (1.62)	341 (1.80)	0.90 (0.70–1.16)	0.77 (0.60–1.01)
>729 days	52 (1.11)	249 (1.32)	0.85 (0.63–1.16)	0.76 (0.55–1.03)
≥300 mg				
<180 days	44 (0.94)	156 (0.82)	1.16 (0.83–1.62)	1.14 (0.81–1.59)
≥180 days	64 (1.36)	378 (2.0)	0.70 (0.53–0.91)	0.61 (0.46–0.81)
180–729 days	40 (0.85)	249 (1.32)	0.84 (0.60–1.19)	0.72 (0.51–1.03)
>729 days	24 (0.51)	156 (0.82)	0.54 (0.35–0.83)	0.48 (0.31–0.75)

* Among current users. ^†^ Odds Ratio (OR) Adjusted only for matching factors (age, sex and calendar year). ^‡^ Odds Ratio (OR) Adjusted for covariates shown in Table 1. Note: Percentages equal to or greater than 2 have been rounded to the first decimal place. Odds ratios and percentages less than 2 have been rounded to the second decimal.

**Table 3 jcm-08-02150-t003:** Sensitivity analysis number 1: using only patients with complete data for serum uric acid (SUA) levels. Risk of AMI among allopurinol current users according to SUA level attained after treatment, duration of treatment, daily dose, and their combinations.

	Cases (%) *n* = 4166	Controls (%) *n* = 16,594	Crude OR ^†^ (95% CI)	Adjusted OR ^‡^ (95% CI)
SUA Level				
<6 mg/dL	70 (1.68)	381 (2.3)	0.74 (0.57–0.96)	0.70 (0.53–0.91)
6–7 mg/dL	45 (1.08)	201 (1.21)	0.91 (0.66–1.26)	0.88 (0.63–1.24)
>7 mg/dL	61 (1.46)	220 (1.33)	1.12 (0.85–1.50)	1.03 (0.76–1.40)
				*p* for trend = 0.011
SUA level among long-term users *				
<6 mg/dL	61 (1.46)	334 (2.0)	0.74 (0.56–0.98)	0.69 (0.51–0.92)
6–7 mg/dL	35 (0.84)	172 (1.04)	0.83 (0.57–1.19)	0.79 (0.54–1.15)
>7 mg/dL	46 (1.10)	148 (0.89)	1.26 (0.90–1.76)	1.11 (0.78–1.58)
				*p* for trend = 0.007
SUA level among long-term users by daily dose *				
<6 mg/dL				
<300 mg	30 (0.72)	159 (0.96)	0.76 (0.51–1.12)	0.69 (0.46–1.04)
≥300 mg	31 (0.74)	175 (1.05)	0.72 (0.49–1.06)	0.68 (0.45–1.01)
6–7 mg/dL				
<300 mg	28 (0.67)	130 (0.78)	0.87 (0.58–1.32)	0.83 (0.54–1.26)
≥300 mg	7 (0.17)	42 (0.25)	0.68 (0.31–1.52)	0.66 (0.29–1.51)
>7 mg/dL				
<300 mg	38 (0.91)	103 (0.62)	1.50 (1.03–2.18)	1.31 (0.88–1.93)
≥300 mg	8 (0.19)	45 (0.27)	0.72 (0.34–1.53)	0.65 (0.30–1.42)

* Among current users. ^†^ Adjusted Odds Ratio (OR) only for matching factors (age, sex and calendar year). ^‡^ Adjusted Odds Ratio (OR) for covariates shown in Table 1. Note: Percentages equal to or greater than 2 have been rounded to the first decimal place. Odds ratios and percentages less than 2 have been rounded to the second decimal. Long-term means 180 days or longer.

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
