# Peer review of "Risk of Acute Myocardial Infarction Among New Users of Allopurinol According to Serum Urate Level: A Nested Case-Control Study"

_jcm, 2019, doi:10.3390/jcm8122150_

Round 1

Reviewer 1 Report

Thanks for clarifying my concerns. 

Reviewer 2 Report

The authors have addressed all comments and remarks in the previous review of the manuscript. There are no additional comments currently.

This manuscript is a resubmission of an earlier submission. The following is a list of the peer review reports and author responses from that submission.

Round 1

Reviewer 1 Report

In this study authors have analysed ~13 years data and have tested the efficacy and role of Allopurinol in reducing the risk of AMI and its cardioprotective effects in hyperuricaemic patients and further assessed if the effect was dependent on dose, duration and serum uric acid (SUA) level attained after treatment. Although the study has been conducted in a well fashioned manner, there are few points that need to clarified.

Major points:

One of the clarification and modification needed in this study is in Figure 1. Figure 1 show that authors have analyzed 4697 patients and 18919 controls and therefore from the figure it looks that sample number is very large. On the contrary Table 2 showed that 86.4% of cases and 85.2% of controls did not use Allopurinol which brings down the number of patients to 465 and controls to 2036. For the benefit of readers, these numbers should also be reflected in Figure 1. Use of statins and drugs blocking the Renin-Angiotensin system have been shown to lower the risk of AMI and authors have shown in Fig 3 that use of either of these along with Allopurinol had no further effects on AMI. One big question that the reader will be asking is that why were statins or other similar drugs administered in these patients? This is not mentioned in any paret of the manuscript. Were these patients hyperlipidemic or were there any other concerns. This need to be clarified as this is AMI eccentric article.

Minor points:

Page 2, line 57 needs saying, “precludes to draw a firm conclusion’ needs to be reworded and softened. Page9, line 235. Is “HR” right, if yes then it needs be defined as this is the first time it is being used in the article. Page 9, Table 5. Under SUA level, <6mg/d shown be written as <6mg/dl. Page 10, Line 278. (since 300 mg) is wrong wording should be replaced with at or above. Line 294 – 297, Page 10 needs to be clarified in a better way and should be re-written.

Reviewer 2 Report

The manuscript by Rodríguez-Martín underlines the importance of consequent therapy of gout with respective therapy, in this case with allopurinol. The reported effect shows the additional use in patients with acute myocardial infarction and underlying elevated levels of uric acid. In these patients medication with allopurinol and reaching a certain threshold of uric acid below 7mg/dL had a lower prevalence of myocardial infarction.

The comments regarding the manuscript are as follows:

The authors should as well describe the risk for the other classical risk factors like for example dyslipidemia, arterial hypertension, diabetes. Which HR was calculated for these factors? In addition, hyperuricemia is associated with cardiovascular disease in general. Did the patients had as well more cerebrovascular accidents in terms of stroke? In the context of a vast body of literature regarding hyperuricemia and consequent lowering the authors should point out which factors are truly novel? Was the effect of lower levels of uric acid in all patients due to treatment with allopurinol alone? Although the study was done in patients with new prescription of allopurinol additional factors might be responsible for the lowering of uric acid.

Reviewer 3 Report

this manuscript is very hard to read and understand. most importantly, the authors need to represent data in the form of figures, not only in tables. it is very hard to find correlations based on numbers from the tables.

Reviewer 4 Report

In this manuscript, Sara Rodriguez-Martin et al. aimed to determine whether allopurinol reduces the risk of acute myocardial infarction (AMI) in patients with gout or asymptomatic hyperuricemia and to assess whether the effect was dependent on dose, duration and serum uric acid level. In general, this retrospective observational study confirmed the protective effects of allopurinol reported previously by different groups.  Here are the concerns by this reviewer:

The major concern is the novelty of this study. As the author mentioned in the introduction, they have previously reported a risk reduction of AMI, dose- and duration-dependent, among allopurinol initiators (in Ref 11). Thus, the scope of this study seems to be the validation of the protective effects of allopurinol which has been observed by other groups as well. In addition, there have been also null or negative results reported previously (Ref 12-15), while the authors did not provide new evidences, especially from mechanism perspective, to thoroughly discuss the controversial observations. Next, as the author mentioned, the relation found between AMI risk and SUA level reached after allopurinol treatment might be a novel point. However, the results were less convinced by lack of a clear definition of the rule for setting the SUA intervals and a control group which treated by other urate-lowering therapy other than allopurinol. In method 2.5, there were only 3 patients with doses over 300mg of allopurinol. However, it was confused that why there were more than 3 patients shown in the Tables 2-5 in 300+ mg, if the number in the table means the number of patients meet each category?

Minor issues:

# was the symbol for co-first authors not *, in line 22.